# Vegetation-based climate mitigation in a warmer and greener World

Ramdane Alkama [1✉], Giovanni Forzieri [1], Gregory Duveiller [1,3], Giacomo Grassi[1], Shunlin Liang[2] & Alessandro Cescatti[1]

The mitigation potential of vegetation-driven biophysical effects is strongly influenced by the background climate and will therefore be influenced by global warming. Based on an ensemble of remote sensing datasets, here we first estimate the temperature sensitivities to changes in leaf area over the period 2003–2014 as a function of key environmental drivers. These sensitivities are then used to predict temperature changes induced by future leaf area dynamics under four scenarios. Results show that by 2100, under high-emission scenario, greening will likely mitigate land warming by $0.71 \pm 0.40\,°C$, and 83% of such effect $(0.59 \pm 0.41\,°C)$ is driven by the increase in plant carbon sequestration, while the remaining cooling $(0.12 \pm 0.05\,°C)$ is due to biophysical land-atmosphere interactions. In addition, our results show a large potential of vegetation to reduce future land warming in the very-stringent scenario $(35 \pm 20\%$ of the overall warming signal), whereas this effect is limited to $11 \pm 6\%$ under the high-emission scenario.

[1] European Commission, Joint Research Centre, Ispra, Italy. [2] Department of Geographical Sciences, University of Maryland, College Park, MD 20742, USA. [3] Present address: Max Planck Institute for Biogeochemistry, Jena, Germany. ✉email: ram.alkama@hotmail.fr

Earth system models (ESMs) project a progressive increase in leaf area index (LAI; the amount of leaf area per unit of ground area) in a large part of the planet over the 21st century[1]. This emerging greening signal has also been detected from satellites in the last three and half decades and attributed to the increase in atmospheric $CO_2$, nitrogen deposition, climate change, and land cover change[2]. Variations in plant physiology, phenology, and structure associated with the greening of the Earth are affecting surface temperatures by altering the water and energy exchanges between land and atmosphere[3–5]. This type of plant biophysics impact on climate is increasingly recognized, given its potential role in enhancing or counteracting the climate benefits of land-based carbon sequestration[6–9]. On the other hand, such effects of vegetation are largely ignored in climate treaties because biophysical effects are uncertain and inconsistent on their sign and magnitude[10–13]. For instance, the resulting net warming or cooling effect of greening patterns largely depends on the climate background, generally leading to local warming at northern latitudes and cooling in tropical and temperate regions[3,14]. An additional layer of complexity to this over-simplified scheme is represented by the projected increase of vegetation density[1] and the concurrent changes in climate, which could potentially amplify or dampen such land-atmosphere interactions and substantially change the sign and magnitude of the net global biophysical effect in the coming decades[3,14].

Given the increasing relevance of nature-based solutions in countries' pledges to meet climate targets[15], it is of foremost importance to quantify in a robust and credible manner the evolution of biophysical and biochemical mitigation potentials of vegetation under future climate conditions. Previous assessments suggest that the ongoing increasing trend in LAI contributed to an overall evaporation-driven cooling effect, particularly pronounced in water-limited environments like the arid Tropics[3,4,16]. Conversely, observation-derived estimates indicate that increases in LAI led to a reduction in surface albedo over boreal areas with extended snow cover, ultimately resulting in a local warming signal[3]. However, the net effect of vegetation and snow cover changes in boreal regions is still controversial[17,18] as model simulations have shown an opposite signal[4]. Besides these differences, observational evidence and model experiments agree on the key role that background climate conditions play in the modulation of biophysical processes mediated by vegetation and, in particular, on the relative importance of radiative versus non-radiative processes[3,14]. The projected decline of key environmental drivers, like snow cover and soil moisture, are therefore expected to influence substantially such land-atmosphere interactions as we move into future climate conditions[19].

Based on the current understanding of the phenomena, it can be speculated that the progressive warming of climate should lead to an enhancement of non-radiative biophysical effects over an increasing share of the Earth[20], thus resulting in amplified mitigation associated with vegetation greening. However, the concomitant rise in atmospheric $CO_2$ concentration could play an opposite influence on land biophysics by reducing water loss during transpiration and producing an increase in the ratio of carbon gain to water loss (i.e. water-use efficiency, WUE)[21,22] via partial closure of stomata[23]. The expected rise in WUE could dampen—or even offset—the cooling associated with the increases in evaporative surfaces related to the greening, and the changes in climate leading to an increase in vapour pressure deficits. Furthermore, robust experimental evidence on the sensitivities of biophysical effects from the background climate are not available yet to support predictions on how these effects will evolve in the future and to improve their uncertain representation in dynamic vegetation models[24]. Due to these complex and contrasting processes and the weaknesses of current vegetation models, the net effect of future combined changes in vegetation density and climate is still rather uncertain and yet to be quantified in a robust manner.

In this study, by fusing Earth observations and Earth system modelling, our analysis shows that the mitigation potential of vegetation-based solutions (afforestation, reforestation, and forest restoration) increases in absolute term, thanks to the amplifying effect of concurrent climate change and increasing plant $CO_2$ sequestration, but declines in relative magnitude, compared to the overall future warming, especially under the more extreme warming scenarios. Half of the biophysical mitigation effect is due to the forecasted increase in vegetation density. The other half is driven by changes in the background climate that amplifies the mitigation potentials of vegetation, thanks to the reduction of radiative warming, mediated by the decrease of snow cover, and the parallel enhancement of non-radiative cooling, due to the increase of evapotranspiration. Altogether, we expect our analysis to add significant value in the ongoing discussion about the role of vegetation on the future climate trajectories, because it provides novel results that heavily rely on Earth observations.

## Results

To address the knowledge gap discussed in the introduction, here we use a combination of Earth observations and Earth system modelling to investigate the global biophysical impacts of future changes in LAI on surface temperature (T) under different scenarios of climate warming and atmospheric $CO_2$ concentration. For this purpose, we first use satellite retrievals to quantify the baseline (2003-2014) monthly sensitivity of T to LAI changes (see methods and data section), as a function of the concurrent variations in snow cover, solar radiation, and evaporation rates. This baseline starts in 2003, which corresponds to the year of the first complete MODIS AQUA records of land surface temperature, used to derive air temperature in this study[25], and ends in 2014, the last year of the historical Coupled Model Intercomparison Project 6 (CMIP6)[26] simulations. We express the sensitivity $dT/dLAI$ as a function of evaporation, solar radiation, and snow cover in order to account for future adaptation of the ecosystem. We have to note here that future evaporation comes from CMIP6 simulations that already account for the adaptation of plants to a changing climate, as for example to an increasing atmospheric $CO_2$.

**Historical sensitivity $dT/dLAI$.** Results show that the sensitivity of T to LAI changes in the presence of snow is dominated by radiative processes driven by the contrast in albedo between vegetation and snow, with a signal that is switching from cooling to warming with the increase of the snow cover fraction (Fig. 1a). In the absence of snow, non-radiative effects mediated by evaporative cooling are dominating the process, with higher sensitivity in arid regions (i.e. with low evaporation rate) (Fig. 1a, b). As expected, the sensitivity $dT/dLAI$ for both radiative and non-radiative processes is enhanced at high radiation levels since it is controlled by the partitioning of solar radiation at the surface.

In order to map the climate sensitivities of biophysical effects from Earth observations, the large scale climate signal on surface temperature first needs to be disentangled from the local effect of vegetation dynamics, following a methodology designed to assess the climate impacts of land cover change[27]. To this aim, for every grid cell, we identified adjacent areas within a 50 km radius where the vegetation cover is stable (i.e. showing less than 0.1 m$^2$/m$^2$ variation in LAI during the observation period). Temporal changes in temperature observed at these reference areas can be fully attributed to climate variability since the contribution of greening is negligible. The variations in T due to LAI changes are then obtained by subtracting the large-scale T signal derived from the reference areas

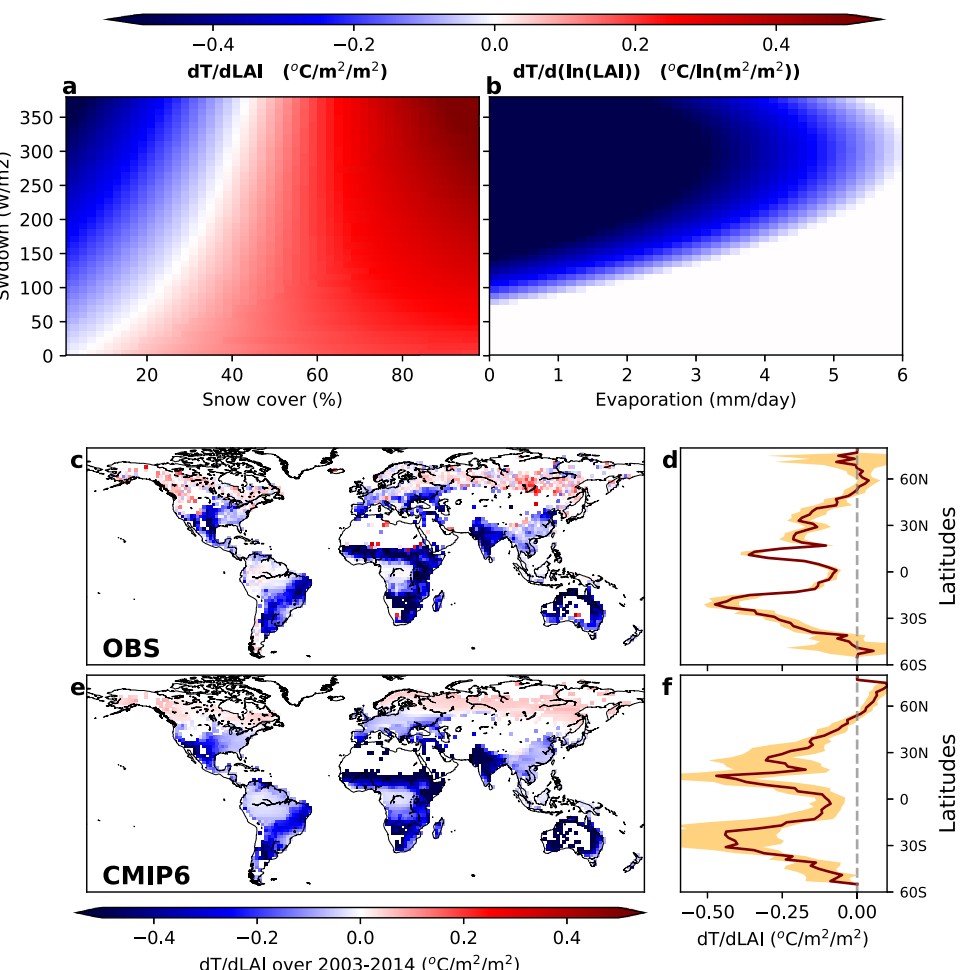

**Fig. 1 Temperature sensitivity to leaf area index over 2003-2014.** Temperature sensitivity to LAI ($ln(LAI)$ in b, where ln is Napierian logarithm) derived from satellite retrievals as a function of downwelling solar radiation ($SWdown$) and **a** snow cover (where and when snow cover exists) and **b** land evaporation (in absence of snow cover). Mean annual sensitivities derived from Earth observations (OBS) by using all combinations of pairs of years in 2003–2014 for each month of the year **c** and median (solid line) of zonal mean ± first and third quartile from the ensemble of years couples **d**. **e** Estimated T sensitivity to LAI for 18 CMIP6 climate models based on modelled values of the drivers (LAI, solar radiation, evaporation and snow cover) and the climate sensitivities derived from space observation in **a**, **b**, while **f** shows the median (solid line) and the min and max zonal mean of the model ensemble (orange envelope).

from the total signal observed at the target grid cell, where both climate variability and vegetation dynamics are at play. The remaining signal reflects the unidirectional control of vegetation on surface temperature for the baseline period (Fig. 1c) and show a clear pattern of warming at northern latitudes and cooling elsewhere, with the largest values in Tropical arid regions, consistently with the results of previous studies[3,4,28].

As a second step, we applied the observed sensitivity $dT/dLAI$ shown in Fig. 1a, b to snow cover, solar radiation, and evapotranspiration predicted by an ensemble of historical simulations (2003–2014) from CMIP6[26]. The resulting map shows that $dT/dLAI$ based on CMIP6 simulations (Fig. 1e, f) compares well ($r = 0.78$, RMSE = 0.13, Supplementary Fig. 1) with observations in terms of both spatial patterns and zonal means (Fig. 1c, d), thereby giving confidence to the application of model outputs for the predictions of future trends in land biophysical processes.

**Future $dT$ under four SSPs scenarios**. To assess the future evolution of the biophysical mitigation driven by vegetation, we combined the data-driven estimates of $dT/dLAI$ with different trajectories of vegetation, snow cover, solar radiation, and evapotranspiration,

consistent with four Shared Socioeconomic Pathways (SSP126, SSP245, SSP370, and SSP585)[29] simulated from 2015 up to the year 2100 from an ensemble of CMIP6 ESMs[26].

All these simulations show a significant increase in LAI by 2100. Moderate increases are simulated under the SSP126 scenario, which depicts the most ambitious mitigation plan (Supplementary Fig. 2a, b). The largest increase in LAI is simulated under SSP585, which assumes continuity of the high-emission scenario (Fig. 2a, b). We have to note here that future SSPs scenarios with larger increases in atmospheric $CO_2$ show larger LAI trends, thanks to both the direct $CO_2$ fertilization effect[2,30] and the indirect effects of warming over Boreal regions[31,32]. The LAI increase is generally associated with an increase in evaporation, except over tropical arid and Mediterranean regions (Fig. 2c, d), where evaporation is limited by water scarcity[33,34], and in the Amazon, likely due to scenarios of widespread deforestation[35]. These climate scenarios also predict a reduction in snow cover that, consistently with warming levels, are strongest in the SSP585 scenario (Fig. 2e, f), followed by SSP370 (Supplementary Fig. 2e, f), SSP245 (Supplementary Fig. 3e, f), and SSP126 (Supplementary Fig. 4e, f). Climate models show a future increase in solar radiation everywhere in SSP126 (Supplementary Fig. 4g,h) and SSP245 (Supplementary Fig. 3g, h), while a decrease

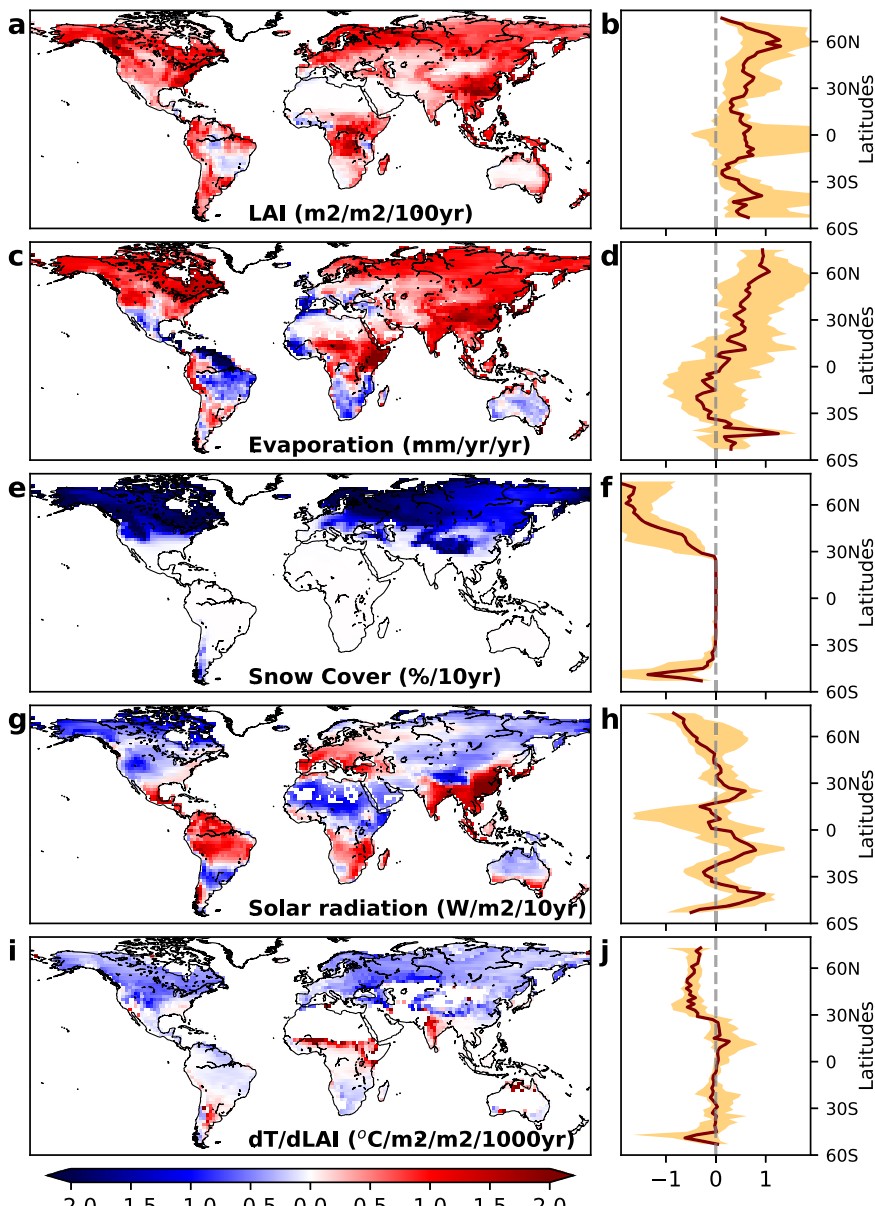

**Fig. 2 Trends derived from an ensemble of 18 climate models over 2015-2100.** Trends in LAI, evaporation, snow cover averaged over the year, surface downwelling shortwave solar radiation (*SWdown*) and *dT/dLAI* from and ensemble of 18 Coupled Model Intercomparison Project 6 (CMIP6) models under Shared Socioeconomic Pathways SSP585 scenario. The median is shown in the left panel (**a**, **c**, **e**, **g**, **i**) while the right panels (**b**, **d**, **f**, **h**, **j**) shows the median (solid line) and the min and max zonal mean of the model ensemble (orange envelope).

in boreal and arid regions are simulated in SSP370 (Supplementary Fig. 2g, h) and SSP585 (Fig. 2g, h). As a consequence of future change in background climate conditions, the variation in *dT/dLAI* sensitivity shows a larger change in the boreal zone (Fig. 2i, j). Indeed, the large decrease in *dT/dLAI* is linked to the reduction in snow cover. Elsewhere, the trend in the sensitivity is more complex. For example, Fig. 1 shows clearly that areas with higher evaporation, such as the tropical rainforests in the Amazonian basin, experience lower *dT/dLAI* compared to vegetation in a dry climate. This may explain why in India, parts of Africa and South America, the sensitivity is getting weaker (Fig. 2i), but the evaporation is getting stronger.

Spatially speaking, the increased *dT/dLAI* is mostly observed in the regions where evaporation is increasing, while decreases in *dT/dLAI* are linked to the reduction in snow cover and/or evaporation.

In response to the projected LAI dynamics, all scenarios show a progressive cooling that is larger for the case with stronger greening trends (SSP585, Fig. 3). Spatially, limited changes in T occur over tropical humid regions, likely due to the low *dT/dLAI* sensitivity in well-watered, densely vegetated canopies that are characterized by high evapotranspiration rates (Fig. 1b). On the contrary, greening drylands shows the largest reduction in T due to the high sensitivity of *dT/dLAI* in condition of high radiation and low evaporation rates (Fig. 1b). At the seasonal scale, Fig. 3b, d, f, h shows maximum cooling during summer, especially over high latitudes. At northern latitudes and for the SSP85 scenario, this temperature mitigation induced by greening between 2015 and 2100 reaches up to 0.55 ± 0.1 °C during the boreal summer. However, this large summer cooling is partially compensated by a slight warming effect during boreal winter due to radiative effect of winter greening over high latitudes.

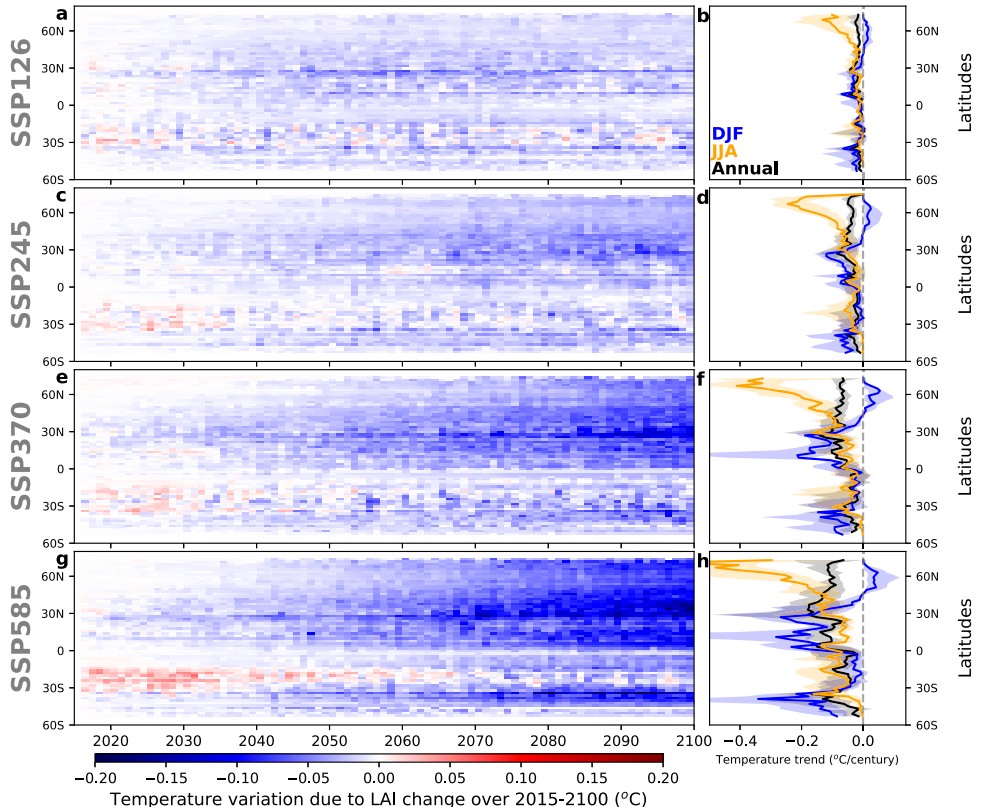

**Fig. 3 Temporal and spatial trend of the latitudinal pattern over 2015–2100.** Temporal and spatial trend of the latitudinal pattern of the land temperature variations (°C) induced by the biophysical effect of LAI change under the four future Shared Socioeconomic Pathways (SSP126, SSP245, SSP370, and SSP585) scenarios: (aceg) time series of annual zonal means, (bdfh) seasonal (December-January-February DJF and June-July-August JJA) and annual latitudinal mean of temperature trends.

Future greening under the SSP585 scenario has a biophysical cooling effect almost everywhere, with a larger magnitude over African savanna (Fig. 4a). Half of this cooling effect is driven by the greening itself (case with future LAI and current climate conditions, Fig. 4c), while the other half is due to the synergic changes in future climate conditions that amplify the biophysical cooling (estimated from the difference between "all effects" minus "LAI effect" Fig. 4b method and data section 7).

These results highlight how the predicted changes in climate will amplify the cooling effect of land greening in all climate zones for two different mechanisms, both leading to the enhancement of biophysical land mitigation (Fig. 2b, d, e, f, g). The first mechanism is the future reduction of snow cover, which will reduce the radiative warming of plant canopies during the winter/spring in the Boreal, Temperate and cold Arid zones, driven by the low albedo of green canopies compared to snow-covered fields (Fig. 4d, e, f). The second mechanism is linked to the enhancement of non-radiative cooling[20] driven by the stronger coupling between LAI and evaporation under future climate condition of increased temperature and atmospheric evaporative demands. At the global scale, future changes in evaporation and snow cover will contribute about equally to the enhancement of vegetation-based biophysical cooling. These mechanisms of actions are also valid for the more ambitious mitigations scenarios, such as SSP126, SSP245 and SSP370 (Supplementary Figs. 5, 6 and 7), albeit with a lower magnitude. We have to note here that, for better consistency between the two hemispheres, we shifted by 6 months the seasonal cycle of the southern hemisphere in Fig. 4d, e, f, g, so that seasonality matches with the solar cycle of the northern hemisphere.

## Discussion
The authors acknowledge the limitations of this study, especially the fact that the non-local biophysical effects driven by large-scale teleconnections are probably not fully captured by our method, despite the role that they can play in the case of large-scale land cover changes[36–38]. Non-local effects are known to originate mainly from changes in large scale circulation that impacts on cloud cover, precipitation and ultimately on incoming solar radiation, snow cover and evaporation. In our analysis, by expressing the sensitivity as a function of these latter three climate drives, we somehow accounted for non-local effects mediated by changes in these variables. However, we estimated the sensitivity of air temperature to LAI over a relatively short time period (2003–2014), thus with limited changes in LAI that were likely not sufficient to trigger relevant non-local effects. Another limitation of this study may come from the use of passive satellite observations at high latitude in the winter. Indeed, additional uncertainties may be in these regions caused by a lack of daylight hours and high solar zenith angles during this period. In order to increase the robustness of our estimates, we used all possible combinations between two years from 2003 to 2014 period, resulting in 66 paired samples. We then excluded grid cells with standard deviation of $dT/dLAI$, from the 66 samples, larger than 0.2. Despite that, Fig. 1d shows large spread in high latitudes variations across longitudes but the general patterns still similar to the two other products (GLOBMAP and COPERNICUS, Supplementary Figs. 13 and 14). In addition, CMIP6 models do not simulate natural shifts in plant species due to climate change which may introduce uncertainties on predicted LAI.

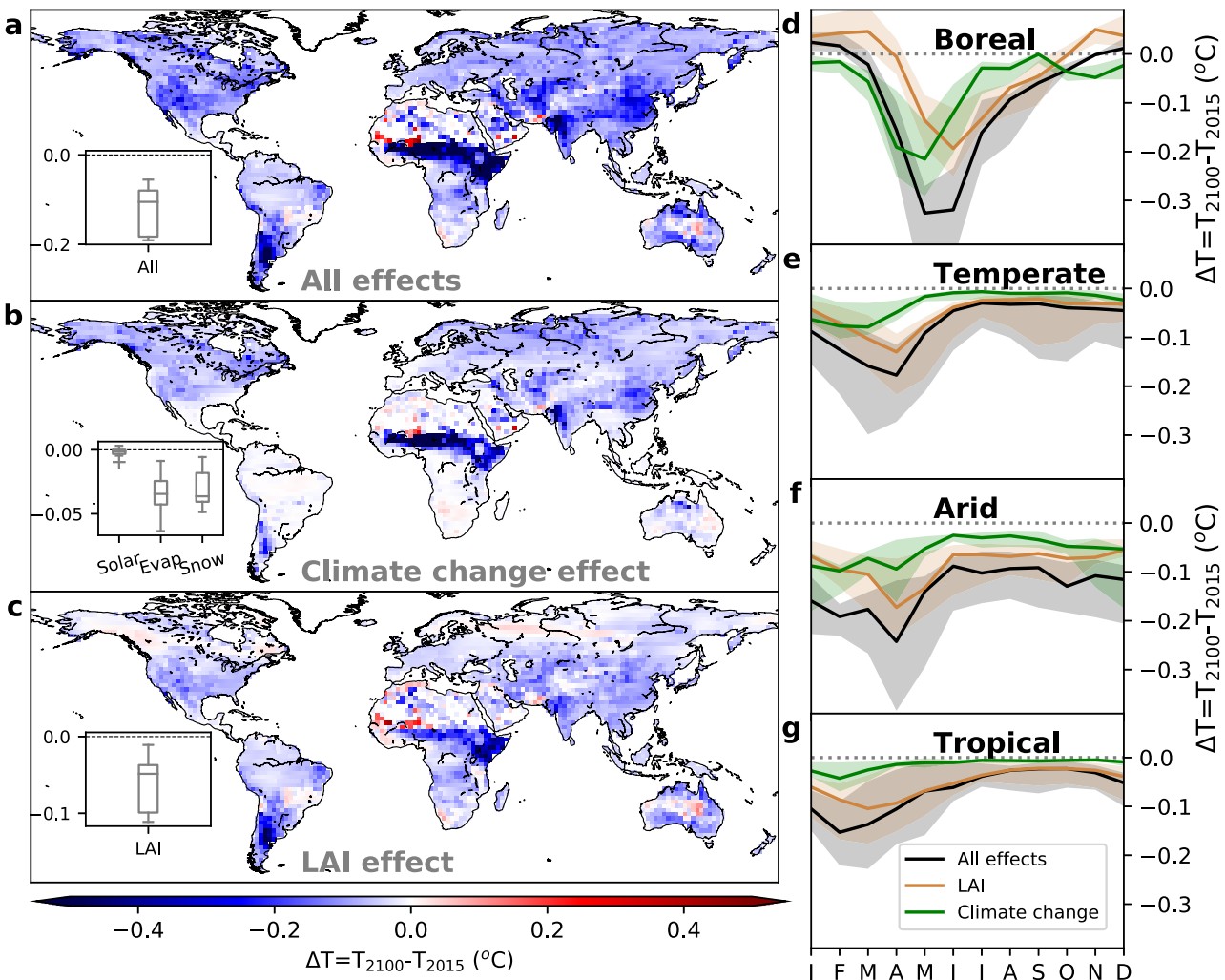

**Fig. 4 Mean annual and seasonal cycle of averaged temperature change over 2015-2100.** Mean annual and seasonal cycle of averaged temperature change over the four climate zones induced by the combined change in LAI and climate under Shared Socioeconomic Pathways SSP585 scenario. The median of Coupled Model Intercomparison Project 6 (CMIP6) ensembles is shown in **a**, **b**, and **c** while the distribution of the entire CMIP6 ensemble is shown in the boxplots, in which the box represents the first, second (median) and third quartiles (whiskers indicate the 99% confidence interval). Black lines in **d**, **e**, **f**, and **g** are the medians of CMIP6 model ensemble when accounting for both LAI and climate change (All effects), whereas brown lines represent the impact of future greening under current climate conditions (LAI effect). The impact of changing the climate conditions at constant LAI is shown by the green line (Climate change effect). Grey, green and brown envelopes show the min and max temperature change in the CMIP6 model ensemble. In the All effects b, the impact of incoming solar radiation (Solar), evaporation (Evap), and snow cover (Snow) are shown by boxplots.

We have to note here that the future projection of LAI increases may be overstated in ESMs[39–41]. This would result in an increasingly warmer world with slowing LAI increases and increasing moisture constraints. In such a case, the medians shown in Fig. 5a are probably an overestimation, and models showing lower mitigation estimates are probably a better approximation of the truth.

In conclusion, we believe that the robustness of our results, which are stemming from evidence-driven analysis, is substantially improving the understanding of present and future biophysical climate impacts of vegetation dynamics under scenarios of combined Earth greening and warming.

Despite the limited mitigation achievable with the predicted LAI dynamics, it is important to consider that the increasing climate mitigation potentials of vegetation may become more relevant if supported by afforestation and restoration programs that may increase the area and density of vegetation. In addition to the biophysical mitigation effects described and quantified before, increasing plants density may also mitigate climate

warming via absorbing some of the $CO_2$ emitted by fossil fuels into the atmosphere. Such effects are known as biochemical effects. Biophysical effects take are mainly driven by the interactions between leaves (LAI) and the atmosphere, while biochemical effects depend mostly in changes in vegetation biomass, which is dominated by woody biomass. This is the reason why we used LAI to study the biophysical effect, while we use total vegetation carbon instead to assess the biochemical effect. Overall, our analysis shows that the mitigation of biochemical processes related to the greening of the Earth (Fig. 5b, method and data section 5) is about five times larger than the biophysical mitigation (Fig. 5a, b). However, the magnitude of biophysical mitigation effects due to greening has a heterogeneous pattern, depending on the combined dynamics of LAI and background climate. Remarkably, the biophysical mitigation signal maximizes during the warmest months and in arid regions, when and where future warming will likely be more challenging for society. In absolute values, both biophysical and biochemical processes driven by LAI show a larger cooling effect in the warmer SSPs,

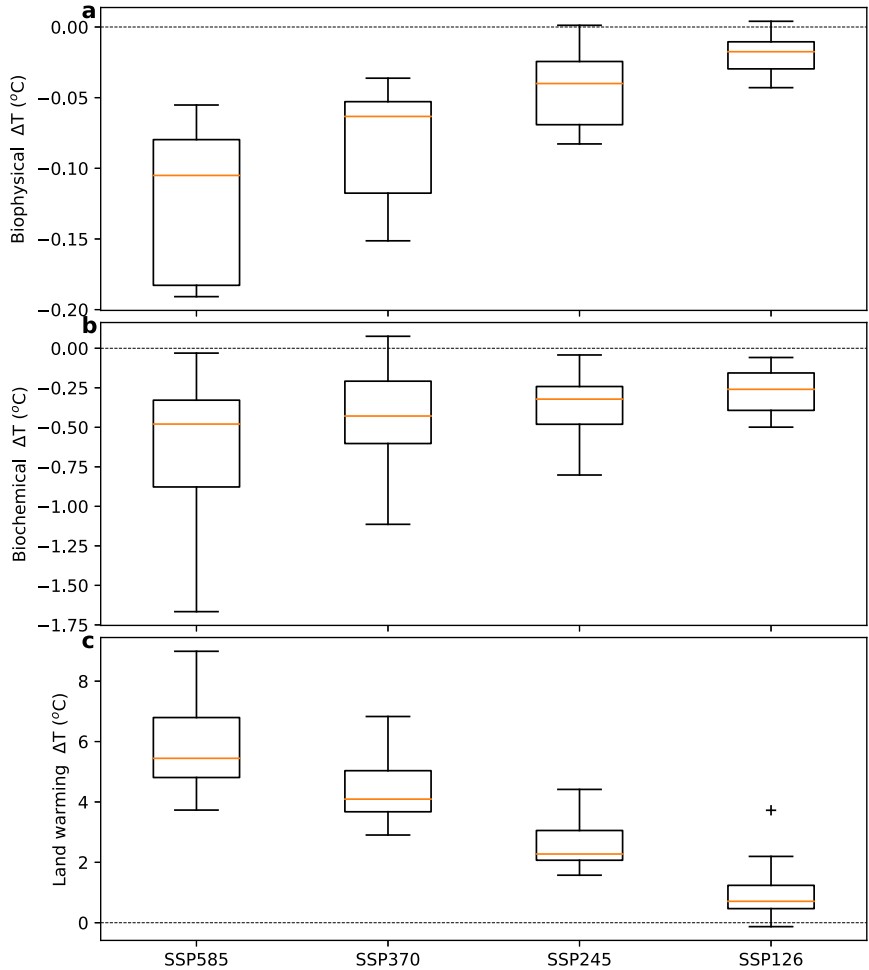

**Fig. 5 Mean land annual temperature change over 2015–2100.** Mean land annual temperature change ($\Delta T$) under the four future Shared Socioeconomic Pathways (SSP126, SSP245, SSP370 and SSP585) scenarios **c** and their associated vegetation mitigation effects due to biophysical **a** and biochemical **b** processes. The Coupled Model Intercomparison Project 6 (CMIP6) simulations are represented by the grey box plots, in which the box represents the first, second (median) and third quartiles (whiskers indicate the 99% confidence interval and grey markers (plus) show outliers).

characterized by higher atmospheric $CO_2$ concentration. However, in relative (compared to global land warming) rate (Fig. 5c), the opposite is found with a larger relative importance of land-based climate mitigation for the more ambitious mitigation plans. Ultimately, these results suggest that the mitigation driven by the greening of the Earth may play an important role in the portfolio of actions to achieve the most ambitious climate mitigation targets.

## Methods

To estimate the plant biophysics in response to the future climate we first estimate the temperature sensitivity to LAI from observations over 2003–2014 under different combination of key environmental drivers such as snow cover, solar radiation, and evaporation. Such datasets are used at 0.05 degree spatial resolution. We then used this sensitivity with future LAI values and climate conditions as simulated by an ensemble of CMIP6 experiments under different SSPs at their common 2 × 2 degree spatial resolution, in order to estimate the future evolution of plant biophysical impacts on climate. To account for the relationship between water use efficiency and atmospheric $CO_2$ concentration, we used evaporation rates rather than soil moisture as a driver of $dT/dLAI$. In fact, plant evaporation simulated in climate models already includes the effect of $CO_2$ fertilization on stomatal conductance.

**Estimation of the local biophysical variation in land air surface temperature due to LAI change.** The goal of this part of the work is to quantify the local climate impacts of observed changes in vegetation density over the period 2003–2014, for which combined observation of leaf area index (GLASS[42]) and air temperature (inferred from satellite observations by Hooker et al. 2018[25]) are available. For this

purpose, the variation of air surface temperatures induced by the change in LAI has been factored out from the natural climate variability using the temperature signal from neighbouring areas with stable LAI.

Our methodology is similar to the one presented in Alkama and Cescatti 2016[43], based on Eq. 1, which assumes that, for a given grid cell, the difference in temperature between two years is equal to the sum of the temperature variation induced by LAI change ($\Delta T_{lai}$) plus the residual signal ($\Delta T_{res}$) due to the natural inter-annual climate variability.

$$\Delta T = \Delta T_{lai} + \Delta T_{res} ==> \Delta T_{lai} = \Delta T - \Delta T_{res} \tag{1}$$

From Eq. 1 it follows that the land use signal $\Delta T_{lai}$ can be quantified as the difference between the observed temperature variations ($\Delta T$) and the natural climate variability ($\Delta T_{res}$). This latter term is estimated from nearby reference grid cells located within 50 km distance ($d_k$) and with stable LAI (i.e. less than 0.1 m²/m² variation in LAI during the observation period). For these grid cells, we can assume $\Delta T_{lai} \simeq 0$ and consequently $\Delta T \simeq \Delta T_{res}$. The residual temperature signal ($\Delta T_{res}$) is therefore the temporal variation in temperature observed in the surroundings of the target grid cell in areas with stable land cover. We used an inverse distance weighting to estimate $\Delta T_{res}$ from the $n$ reference grid cells, according to Eq. 2.

$$\Delta T_{res} = \frac{\sum_{k=1}^{n} \frac{\Delta T_k}{d_k}}{\sum_{k=1}^{n} \frac{1}{d_k}} \tag{2}$$

From Eqs. 1 and 2, we estimate $\Delta T_{lai}$ for all grid cells where we observe LAI change larger than 0.1 m²/m². This calculation of these differences is repeated between two years for each individual month, for all the 66 pairs of years available for the period 2003–2014.

**Sensitivity of air surface temperature to LAI change under climate conditions.** Once the monthly local sensitivity is estimated as described above (section 1.1), we

split the data in two regions (with and without snow using 1% monthly snow cover as threshold). The snow-covered areas are known to be dominated by the radiative effect (i.e. due to the contrast in albedo between vegetation and snow), while the snow-free areas are dominated by the partitioning of the available energy in turbulent fluxes[44] (i.e. evaporation versus sensible heat). The sensitivity $dT/dLAI$ in the first areas are then expressed as a function of monthly snow cover ($SC$ in %) estimates from MODIS (MYD10CM.006) and solar radiation ($SWdown$ in W/m$^2$) from the ERA5[45] reanalysis using the bivariate quadratic least square regression and the resulted function is as follow.

$$dT/dLAI = -1.66\,10^{-5}SC^2 - 2.14\,10^{-7}SW_{down}^2 + 1.56\,10^{-5}SC\,SW_{down}$$
$$+ 2.9\,10^{-3}SC - 8.23\,10^{-4}SW_{down} - 0.011 \quad (3)$$

Similarly, for snow-free conditions, the sensitivity is formulated as a function of land evaporation (E in mm/day) fluxes coming from Global Land Evaporation Amsterdam Model (GLEAM version 3.1a product[46]) and solar radiation. However, in order to account for the variation of the sensitivity $dT/dLAI$ as a function of absolute LAI (lower sensitivity with high LAI) as shown in Fig. 1c, we used naperian logarithm of LAI to fit our bivariate quadratic function as follow.

$$dT/d(ln(LAI)) = 3.64\,10^{-3}E^2 + 1.17\,10^{-5}SW_{down}^2 - 5.80\,10^{-5}E\,SW_{down}$$
$$+ 8.51\,10^{-2}E - 6.74\,10^{-3}SW_{down} + 0.42 \quad (4)$$

In both Eqs. (3) and (4), the left side "d" represent the difference between two years while the right members of the equation (E, SWdown and SC) are the average between the two years.

We have to note here that this sensitivity (Fig. 1) is computed with the assumption of a 1 m$^2$/m$^2$ LAI change during all months of the year. However, over some climate regions (e.g. the boreal zone), the observed greening is mainly driven by the growing season when snow cover is absent. Consequently, in those regions the annual signal is dominated by summer values and may lead to a mean cooling effect[4,28] despite the winter warming.

**Climate zone map**. In this study, climate zones were defined according to the latest digital Köppen-Geiger World map of climate classification for the second half of the 20th century[47]. This map is based on data sets from the Climatic Research Unit (CRU TS2.1) at the University of East Anglia and the Global Precipitation Climatology Centre (GPCC) at the German Weather Service.

The Köppen-Geiger World map contains 31 climate zones on a regular 0.5 degree lat/lon grid for the period 1951 to 2000 (http://koeppen-geiger.vu-wien.ac.at/present.htm). We merged the 31 climate zones into 5 major zones (Equatorial, Arid, Temperate, Boreal, Polar) as defined in the classification system[47] (Supplementary Fig. 8). The polar zone was not analysed since it does not include relevant vegetated areas. We have to note here, that the climate zones are kept constant when doing statistics by climate zone even for future climate.

**LAI product**. We evaluated five (GLASS[42], GLOBMAP[48], GIMMS3g[49], Copernicus[50], and LTDR[51]) LAI datasets derived from Earth Observations against air surface temperature[25] at monthly time step over 2003–2004 (see section 4). We finally selected the GLASS[42] product for multiple reasons. First, all products, except GLASS and GLOBMAP, are produced using an ensemble of different satellites/sensors in time over 2003–2014, which may introduce inconsistencies in the time series[52]. Direct validation to ground LAI observations both globally and over China demonstrates that GLASS LAI shows the best performance over 2000–2017[53]. In addition, GLASS is gap-free and shows robust $dT/dLAI$ signals (Supplementary Figs. 9 to 12). Copernicus[50] and LTDR[51] LAI were excluded because they present gaps especially in snow-covered regions, which make them inadequate for the assessment of temperature sensitivity in cold climates. GIMMS3g[49] and GLOBMAP[48] were excluded because they show less clear signals compared to GLASS and have a coarser spatial resolution (0.083 and 0.073 degree, respectively) compared to other LAI products (available at 0.05 degree). Thus, interpolating air surface temperature from 0.05 to coarser resolution would have introduced further uncertainties.

Another independent validation study also showed that the GLASS LAI product has the lowest uncertainty, followed by GEOV1 and MODIS for all the biome types tested[54]. The GLASS LAI product is robust, particularly over the snow-covered regions, mainly due to the novel inversion algorithm and surface reflectance preprocessing technique (Liang, et al., DOI:10.1175/BAMS-D-1118-0341.1171). Unlike other methods that use only satellite data acquired at a specific time to retrieve LAI, the GLASS algorithm uses an entire year of surface reflectance to estimate the one-year LAI profile for each pixel. Furthermore, the surface reflectance data from atmospheric correction is frequently contaminated by clouds, the GLASS team uses an effective pre-processing method[55] to generate temporally continuous and smoothed surface reflectance time series in eliminating the impacts of this source of "noise".

To test the robustness of our analysis, we performed additional tests using Copernicus LAI, which is one of the three (GLASS, Copernicus and LTDR) high spatial resolution products, and find similar results (Supplementary Figs. 13–14). We choose to use Copernicus instead of LTDR because this latest shows strange patterns for LAI values around 4.5 m$^2$/m$^2$ (Supplementary Figs. 10-11). We also

tested GLOBMAP at its native spatial resolution and find similar results (Supplementary Figs. 13–14).

Because of the use of neighbouring grid cells, the calculation process is computational demanding. For this reason, we limited the test of the 5 different existing LAI products to two consecutive years (2003–2004) before taking a decision about the LAI product to be used for the whole period 2003–2014.

We then proceeded as follows. For a given area, the $\Delta T_{lai}$ values derived with Eqs. 1 and 2 are summarized in a plot (Supplementary Figs. 10–13) where each cell $i$ shows the average $\Delta T_{lai,i}$ observed for a given combination of LAI in the two observation years (here 2003 and 2004) reported on the X and Y axis, respectively, where the axes are discretized with 1% bins. Supplementary Figs. 9–12 can be interpreted as the difference between similar graphs drawn for the temperature variation ($\Delta T$) and the residual signal ($\Delta T_{res}$) according to Eq. 1. Because of the regular latitude/longitude grid used in the analysis, the area of the grid cells ($a_m$) varies with the latitude. The temperature signal ($\Delta T_{lai,i}$) is therefore computed as an area-weighted average (Eq. 5).

$$\Delta T_{lai,i} = \frac{\sum_{m=1}^{M} a_m \Delta T_{lai,m}}{\sum_{m=1}^{M} a_m} \quad (5)$$

All vegetated grid cells were classified in one of the four major zones according to the Köppen-Geiger classification (Supplementary Fig. 8).

**CMIP6 climate simulations**. Historical (2003-2014) and the four SSPs (SSP126, SSP245, SSP370, and SSP585) scenarios (2015–2100) of simulated LAI, incoming solar radiation, snow cover, evapotranspiration, and vegetation carbon stock coming from 18 (CanESM5, CESM2, CESM2-WACCM, CNRM-ESM2-1, EC-Earth3-Veg, GFDL-CM4, GFDL-ESM4, GISS-E2-1-G-CC, GISS-E2-1-G, GISS-E2-1-H, HadGEM3-GC31-LL, IPSL-CM6A-LR, MIROC6, MIROC-ES2L, MPI-ESM1-2-HR, MRI-ESM2-0, SAM0-UNICON, UKESM1-0-LL) climate models from CMIP6 archive, are used in the current study. The SSPs scenarios are used to derive greenhouse gas emissions scenarios with different climate policies. In the SSP126 the world shifts gradually, but pervasively, toward a more sustainable path. In the SSP245 the world follows a path in which social, economic, and technological trends do not shift markedly from historical patterns. In the SSP370 the economic development is slow, consumption is material-intensive, policies shift over time to become increasingly oriented toward national and regional security issues and inequalities persist or worsen over time. While in the SSP585, we assume the continuity of high emissions. The historical simulations of the CMIP6 archive were used to assess how climate models compare with observations, while the future simulations were used to investigate the future impacts of LAI dynamics on the climate system under different future climate trajectories. All climate models use land cover change scenarios from land use harmonization datasets[56] (https://luh.umd.edu/) and simulate tree mortality and fires. However, natural shifts in plant species due to climate change is not simulated. All CMIP6 model outputs are bilinearly interpolated to a common 2 × 2 degree spatial resolution.

**Biochemical effect of LAI on air temperature**. A previous study[57] shows a strong linear relationship between atmospheric carbon concentration and regional surface air temperature. Here, we combined the strong linearity of the regional climate response over most land regions presented from Leduc et al.[57] and the simulated variation in vegetation carbon stock by CMIP6 climate models, to drive the global-scale biochemical climate impacts. Basically, Leduc et al. 2016 find an increase of land temperature of 2.2 ± 0.5° per 1 Terra ton of carbon (Tt C) in the atmosphere. In our case we used the total increase of carbon in plants ($\Delta B$ in Tt C) between 2015 and 2100 coming from the CMIP6 archive to estimate the biochemical effect as shown by Eq. (6).

$$\Delta T = \frac{2.2\,\Delta B}{1} \quad (6)$$

**Air temperature**. Air temperature product used in the present study is produced by combining MODIS day and night land surface temperature (LST) and observed in-situ air temperature[58] using a statistical model that incorporates information on geographic and climatic similarity. One of the reason for the use of LST day and night is to account for the landscape differences between land cover types. It is, for example, well known that the daily temperature amplitude is lower over dense vegetation compared to lower density or bare soil[27,59,60]. Since many of the meteorological sites used in the development of this product occur in low-elevation, homogenous and developed areas, climatic similarity statistics are done using the WorldClim_v1.4[61] data that is mainly built to overcome this kind of problem by, for example, the use elevation statistics. In addition, in order to account for the climate drivers (eg. solar radiation), the geographically weighted regression is used within the in-situ air temperature. Despite the fact this product uses some complex statistics to overcome the issues described here and also the fact that this product is validated[58] against ERA5 reanalysis data, we cannot exclude that part of the uncertainties found in this study originates from the use of this product itself especially over elevated land where the difference with ERA5 is larger.

**Generating Fig. 4**. "LAI effect" is estimated from the multiplication of current sensitivity $dT/dLAI$ by LAI trend. In "all effects", we first estimate new $dT/dLAI$ from simulated future solar radiation, evaporation and snow cover, and then multiplied the new sensitivity by LAI change. While, "climate change" effect is the difference between the two. The individual terms of climate change are estimated by subtracting "all effects" from "all effect except individual term that was kept constant".

We have to note here that this method of separation is used to have an approximated numbers of each driver. However in the real world the interactions between the different climate drivers makes difficult to separate them precisely.

## Data availability

The data that support the findings of this study are openly available. GLASS LAI is accessible at http://www.glass.umd.edu/index.html. Air temperature is accessible at https://figshare.com/collections/A_global_dataset_of_air_temperature_derived_from_satellite_remote_sensing_and_weather_stations/4081802/1. ERA5 surface incoming solar radiation is accessible at http://climate.copernicus.eu/climate-reanalysis. GLEAM evaporation accessible at https://www.gleam.eu/. MODIS snow cover is accessible at https://nsidc.org/data/MYD10CM/versions/6. CMIP6 simulations are accessible at https://esgf-node.llnl.gov/search/cmip6/. COPERNICUS LAI is accessible at https://land.copernicus.eu/global/products/lai. GLOBMAP LAI is accessible at https://zenodo.org/record/4700264#.YParmqaxVUQ. GIMMS LAI is accessible at http://cliveg.bu.edu/modismisr/lai3g-fpar3g.html. LTDR LAI is accessible via google earth engine at https://developers.google.com/earth-engine/datasets/catalog/NOAA_CDR_AVHRR_LAI_FAPAR_V5#description.

## Code availability

The programs used to generate all the results are made with Python and Fortran. Analysis scripts are available at https://github.com/RamAlkama/FutureVegBioph.

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

## Acknowledgements
We acknowledge the World Climate Research Program's Working Group on Coupled Modeling, which is responsible for CMIP6, and thank the climate modeling groups for producing and making available their model outputs. We also thank the GLASS, GLOBMAP, GIMMS, LTDR, and COPERNICUS groups that produce LAI from satellite products. We thank GLEAM group for producing evaporation product, ERA5 for incoming solar radiation and MODIS for snow cover. The study was funded by the European Commission, Joint Research Centre. However, the views expressed are purely those of the writers and may in no circumstance be regarded as stating an official position of the European Commission.

## Author contributions
R.A. directed this work with contributions from all (A.C., G.F., G.D., G.G., and S.L.) authors. R.A. performed the analysis. All (R.A., A.C., G.F., G.D., G.G., and S.L.) authors discussed the results and contributed to writing the paper.

## Competing interests
The authors declare no competing interests.
