## [Peer Review File · Nature Communications]

REVIEWER COMMENTS

Reviewer #1 (Remarks to the Author):

Overview:

This study aims to quantify surface air temperature (Ts) changes induced by future leaf area index (LAI) dynamics derived from four Shared Socioeconomic Pathways (SSPs) from an ensemble of Coupled Model Intercomparison Project 6 (CMIP6) Earth System Models (ESMs). First, the authors use satellite-based observations to quantify the sensitivity of Ts to changes in LAI as a function of key environmental drivers including solar radiation, evaporation, and snow cover over the period 2003-2014. Satellite-based observations of Ts and LAI were derived from a statistical model based on MODIS Collection 5 land surface temperature (LST) data and the GLASS LAI product based on MODIS Collection 5 surface reflectance (SR) data, respectively. The author next applied these sensitivities to LAI and environmental outputs from four SSPs from an ensemble of CMIP6 ESMs to predict biophysical changes resulting from changes in LAI out to 2100. In response to widespread projected increases in LAI across all models, all scenarios show a progressive biophysical cooling that is positively correlated with greening.

This is an interesting paper on an important mechanism that is missing from ESMs. The research is well executed and the writing is compelling. The work is also of broad importance given that it provides an estimate of the potential impact of biophysical climate regulation associated with changes in LAI. However, in its current form there are too many potential issues with and missed details in the methods section. Most importantly, I feel the role of land cover change and its influence on dTs/dLAI are a missing component that could greatly impact the results and conclusions of the manuscript. Before I can make a recommendation, these issues must be addressed. Please find below my specific comments.

Major Comments:

1. The introduction is very well written and compelling.

2. Ts observations appear derived from a monthly product that integrates meteorological estimates of 2-m air temperature and MODIS Collection 5 LST data (Hooker et al. 2018). Very few details are provided on this product currently. Given the importance of this dataset to the results, at minimum, key details on how this product was derived need to be stated in a short methods section. For instance, how were differences between land cover type, elevation, incoming radiation, air temperature, and surface temperature accounted for when extrapolating beyond meteorological sites? Many of the meteorological sites used in the development of this product occur in low-elevation, homogenous, developed areas, which could pose a problem for accurate extrapolation. Further, MODIS Collection 5 LST data were used and these data have been phased out due to sensor degradation issues. New MODIS Collection 6 LST uses a new algorithm that incorporates key improvements. Are the findings of this study robust to these potential issues?

3. LAI Product: The authors state that they evaluated 5 LAI products but do not explicitly state which ones (Line 322). They then go on to state, "First, all products, except GLASS, are produced using an ensemble of different satellites/sensors in time over 2003-2014, which may introduce inconsistencies in the time series⁴⁶" (Line 324). I take issue with this statement and with the way the author's have evaluated LAI products. First, many products are based on only MODIS surface reflectance observations over the full 2003-2014 period. These are the best available observations over the full time period, and thus it seems most appropriate to limit the LAI product comparison to those that are derived from MODIS surface reflectance if possible. It is my understanding that GLASS and GLOBMAP LAI products are based on MODIS C5 surface reflectance (Jiang et al 2017). Jiang et al 2017 found that the MODIS C6 LAI trend is positive, whereas the GLASS and MODIS C5 LAI trend is negative from 2003-2011 (very similar to the period of study here). This difference is attributed to MODIS sensor degradation, which was corrected in the MODIS C6 product. How does uncertainty in LAI trends impact the findings presented here? I acknowledge that it's very good to see generally consistent relationships between LAI and temperature (Figure S12). That said, I feel it's important that the author's are using sound logic in their LAI product

evaluation and able to show their findings are robust across multiple LAI products.

Jiang C, Ryu Y, Fang H, Myneni R, Claverie M, Zhu Z. 2017. Inconsistencies of interannual variability and trends in long-term satellite leaf area index products. *Global Change Biology* 23: 4133–4146.

4. The loss of snow cover in winter at higher latitudes should result in local warming due to changes in the albedo. However, a main issue with passive satellite observations at high latitude in the winter is a lack of daylight hours and high solar zenith angles. Thus, how confident are the authors in their high-latitude, winter-time estimates of $dT/dLAI$? What steps were taken to account for these issues with satellite observations?

5. Again, very little detail is provided on the input environmental variables used in the analysis. For instance, a single line indicates that evaporation data from the GLEAM model was used in the analysis (Line 301). However, there are multiple GLEAM model versions and data products. There needs to be more detail included here so that the analysis is reproducible. Details could reveal issues with the analysis also. For instance, does this product use satellite LST data as an input? If so there is circularity when comparing changes in LST-based air temperature with LST-based estimates of transpiration.

6. It is also unclear how sensitivity $dTs/dLAI$ was calculated "as a function" of snow cover, ET, and radiation (Lines 303 - 307). Were ET, radiation, and snow explicitly included as variables in a multivariate framework? Additional details are needed before the sensitivity analysis can be fully evaluated.

7. A final potentially major issue with this analysis is that there is no explicit treatment of land cover change. Previous research has found that for key regions much of the trend in LAI is driven by changes in land cover (Wang et al. 2019, 2020). Indeed, land cover change would result in a step change in LAI and LST that would greatly impact $dTs/dLAI$ estimates. It is problematic if these estimates are then applied to extrapolate dT from $dLAI$ in ESMs (especially if these ESMs used fixed land cover maps), which are largely predicting vegetation density change. At minimum, this limitation of the approach needs to be discussed in detail.

Wang JA, Friedl MA. 2019. The role of land cover change in Arctic-Boreal greening and browning trends. *Environmental Research Letters* 14: 125007.

Wang JA, Sulla-Menashe D, Woodcock CE, Sonnentag O, Keeling RF, Friedl MA. 2020. Extensive land cover change across Arctic-Boreal Northwestern North America from disturbance and climate forcing. *Global Change Biology* 26: 807–822.

8. Related to Comment 4, how do the models treat land cover change out to 2100? Are some using static land cover? With climate change, we know there could be rapid shifts in ecosystem composition from the conversion of tropical forests to savannas to the shrubification of the arctic. Given different vegetation function types have different $dTs/dLAI$ due to different life history strategies, I suspect this is a large area of simplification in this study. This study is still very valuable and these complexities are not easily quantified, however, they should be stated as a limitation and discussed.

9. Is it appropriate to extrapolate static Ts sensitivities found here to 2100? The sensitivity estimates are based on 10 years worth of data and it's unrealistic to assume they will remain static over time. We know ecosystems adapt to climate over long time periods. For example, trees might respond to increasing atmospheric CO_2 by modifying key hydraulic traits, which might lead to a change in ET and consequently a change in air temperature, and this could happen independent of change in LAI. There should at minimum be discussion of this simplifying assumption.

10. Additionally, I feel the authors should at least address the case that projected increases in LAI may be overstated in ESMs (Smith et al. 2016; Yuan et al. 2019; Wang et al. 2020). If this is the case and we are moving towards a warmer world with slowing LAI increases and increasing

moisture constraint, could the authors infer from the observational work what might happen with biophysical climate regulation? Could there be a positive feedback in this case? It could be useful to point this potential option out in the discussion as a point of uncertainty and need for future research.

Wang, S., et al. 2020. Recent global decline of CO₂ fertilization effects on vegetation photosynthesis. *Science* 370, 1295-1300.

Yuan, W., Zheng, Y., Piao, S., Ciais, P., Lombardozzi, D., Wang, Y., Ryu, Y., Chen, G., Cox, P., Dong, W., Hu, Z., Jain, A.K., Jiang, C., Kato, E., Li, S., Lienert, S., Liu, S., Nabel, J., Qin, Z., Quine, T., Sitch, S., Smith, W.K., Wang, F., Wu, C., Xiao, Z., Yang, S. 2019. Increased atmospheric vapor pressure deficit reduces global vegetation growth. *Science Advances* 5, eaax1396.

Smith, W.K., Reed, S.C., Ballantyne A.P., Cleveland, C.C., Anderegg, W.R.L., Wieder W.R., Running, S.W. 2016. Large divergence of satellite and Earth system model estimates of global terrestrial CO₂ fertilization. *Nature Climate Change* 6, 306–310.

Minor Comments:

1. Abstract: SSps and CMIP5 are undefined acronyms.
2. Line 301. GLEAM undefined.
3. Line 322: State the five products that were compared up front.
4. Why are Fig. 1a,b cut off at SWdown 350 W/m²?

Reviewer #2 (Remarks to the Author):

Reviews of Manuscript No.: NCOMMS-21-18083-T

Title: Vegetation-based climate mitigation in a warmer and greener World

Author(s): Ramdane Alkama, Giovanni Forzieri, Gregory Duveiller, Giacomo Grassi, Shunlin Liang and Alessandro Cescatti

Overall conclusions and recommendations:

This manuscript tackles an original question regarding the mitigation potential of vegetation-driven biophysical effects under different future climate change scenarios. Authors use remote sensing datasets including air temperature and LAI to estimate the temperature effects in LAI changes. According to the temperature sensitivity, authors further establish a function to explore the temperature sensitivity effected by snow cover, solar radiation and evaporation fluxes. Authors then calculated the temperature changes induced by the LAI altering under four climate change scenarios based on the CMIP6 archive. The main findings including 1) the biochemical effect shows more important than the biophysical effect in the greening world, 2) a large potential of vegetation to reduce future land warming in the most ambitious mitigation plan SSP126 scenario than the business as usual SSP585 scenario.

The study is within the scope of the journal *Nature Communications*, and its topic and conclusions could attract the interest of readers working in areas of climate change mitigation and land use/cover management. This manuscript has a clear structure, the quality of presentation is acceptable. It has potential to be a good reference. In the following, I provide more specific comments (major and minor) and suggestions to improve the manuscript.

Specific comments:

1. Authors introduce a function between temperature changes induced by LAI change and three environmental drivers, solar radiation, evaporation and snow cover. However the function is not clear. Although the manuscript shows the function in Figure 1ab. But it not easy to follow. Please add the equations.
2. The manuscript distinguishes the biophysical and biochemical effect of LAI on air temperature. How to distinguish them?
3. This manuscript introduces multi-datasets including remote sensing datasets, climate modelling datasets from CMIP6 archive. Different data/model have different spatial resolution. How to compare these datasets? Which resolution/grid size is used in the analysis?
4. "Results show that, under the SSP585 scenario, by 2100 greening will likely mitigate land warming by $0.71 \pm 0.40^\circ\text{C}$, and 83% of such effect ($0.59 \pm 0.41^\circ\text{C}$) is driven by the increase in plant carbon sequestration, while the remaining cooling ($0.12 \pm 0.05^\circ\text{C}$) is due to biophysical land-atmosphere interactions." I can not find/read these numbers in any figures.
5. L25, "Half of the biophysical mitigation effect ...", how to quantify the half?
6. L32-37, the sentence is not easy to follow, please revise it.
7. L140, and SSP245 (Fig. S2gh) -> and SSP245 (Fig. S3gh)
8. L300, add reference for ERA5 reanalysis.
9. Figure 5b, y-axis title, Biogeochemical -> Biochemical
Figure 5 is not being cited in the manuscript. Please check it.

Reviewer #3 (Remarks to the Author):

The study by Alkama et al. investigated the biophysical and biochemical impacts of recent and future greening on temperature using observations and CMIP6 simulations. The authors suggested that the Earth's greening can be a potential approach for future climate mitigation due to its cooling effects. Overall, this study is on a topic of relevance and general interest to the readers of Nature Communication. However, the methodology needs some clarification, and the results need more explanations and discussions.

My major concerns are:

1. It is not clear how $dT/dLAI$ is calculated. For instance, it is very confusing to see $dT/dLAI$ and $dT/d\ln(LAI)$ in Figure 1. What is their difference? Regarding Method section 1.2, what is the reason to use snow cover, solar radiation, and ET to describe climate conditions? Why not use temperature and precipitation? "The sensitivity is estimated as a function of monthly snow cover and solar radiation", "the sensitivity is formulated as a function of land evaporation fluxes and solar radiation". Can the authors provide more explanations of their methods here?
2. I like the idea of separating the biophysical and biochemical impacts. However, without reading the paper (Leduc et al 2016), readers cannot understand how the biochemical effects are calculated? As the authors suggested, the biogeochemical feedback of Earth's green is about five

times larger than the biophysical feedback. However, there is no evaluation of the simulated biogeochemical feedback in CMIP6. For instance, can the biochemical effects in CMIP6 be compared with observations like the biophysical effects in Figure 1? Results in Figure 5 are never explained or discussed in the manuscript.

3. In Figure 4, how were the "all effect", "climate change effect (and its individual terms)", "LAI effect" defined and calculated? In the "climate change effect", would the "Evap" effect be considered as the results of the "LAI effect"?

Others:

1. It would be better to use a consistent term: "evaporation" vs. "evapotranspiration"; "biogeochemical" vs. "biochemical"; "biogeophysical" vs. "biophysical"
2. According to Figure 2, the sensitivity ($dT/dLAI$) is getting weaker and ET is getting stronger in India, parts of Africa and South America, where there is high sensitivity (more negative) at present according to Figure 1. First, the statement "the increased sensitivity is mostly observed in the regions where evaporation is increasing" seems incorrect. It is increased $dT/dLAI$, but decreased sensitivity. Second, any explanation about what caused the weaker sensitivity?

Dear Editor,

Please find below the referees comments in black and our answers in blue.

Reviewer #1 (Remarks to the Author):

Overview:

This study aims to quantify surface air temperature (T_s) changes induced by future leaf area index (LAI) dynamics derived from four Shared Socioeconomic Pathways (SSPs) from an ensemble of Coupled Model Intercomparison Project 6 (CMIP6) Earth System Models (ESMs). First, the authors use satellite-based observations to quantify the sensitivity of T_s to changes in LAI as a function of key environmental drivers including solar radiation, evaporation, and snow cover over the period 2003-2014. Satellite-based observations of T_s and LAI were derived from a statistical model based on MODIS Collection 5 land surface temperature (LST) data and the GLASS LAI product based on MODIS Collection 5 surface reflectance (SR) data, respectively. The author next applied these sensitivities to LAI and environmental outputs from four SSPs from an ensemble of CMIP6 ESMs to predict biophysical changes resulting from changes in LAI out to 2100. In response to widespread projected increases in LAI across all models, all scenarios show a progressive biophysical cooling that is positively correlated with greening.

This is an interesting paper on an important mechanism that is missing from ESMs. The research is well executed and the writing is compelling. The work is also of broad importance given that it provides an estimate of the potential impact of biophysical climate regulation associated with changes in LAI. However, in its current form there are too many potential issues with and missed details in the methods section. Most importantly, I feel the role of land cover change and its influence on $dT_s/dLAI$ are a missing component that could greatly impact the results and conclusions of the manuscript. Before I can make a recommendation, these issues must be addressed. Please find below my specific comments.

We thank the reviewer for the positive comments. Just for clarification, in the current study we used GLASS LAI v40 that is based on MODIS collection 6 not 5 as stated by the reviewer. In addition, land cover change may impact $dT_s/dLAI$ mainly by changing evaporation rates or by changing albedo via its impact on snow cover. These two effects are accounted in the current study when expressing $dT_s/dLAI$ as function of evaporation, snow cover and solar radiation.

Major Comments:

1. The introduction is very well written and compelling.

We thank the reviewer for this positive comment

2. T_s observations appear derived from a monthly product that integrates meteorological estimates of 2-m air temperature and MODIS Collection 5 LST data (Hooker et al. 2018). Very few details are provided on this product currently. Given the importance of this dataset to the results, at minimum, key details on how this product was derived need to be stated in a short methods section. For instance, how were

differences between land cover type, elevation, incoming radiation, air temperature, and surface temperature accounted for when extrapolating beyond meteorological sites? Many of the meteorological sites used in the development of this product occur in low-elevation, homogenous, developed areas, which could pose a problem for accurate extrapolation. Further, MODIS Collection 5 LST data were used and these data have been phased out due to sensor degradation issues. New MODIS Collection 6 LST uses a new algorithm that incorporates key improvements. Are the findings of this study robust to these potential issues?

Thanks for pointing this. Ts product was already documented in Hooker et al. 2018 and validated against ERA5 reanalysis data. This is the reason why we did not provide more details on it in the previous version of the manuscript. However, we agree with reviewer on fact that additional information's should be included. "Air temperature product used in the present study is produced by combining MODIS day and night land surface temperature (LST) and observed in-situ air temperature (Hooker *et al.*, 2018) using a statistical model that incorporates information on geographic and climatic similarity. One of the reason of the use of LST day and night is to account for the landscape differences between land cover types. It is, for example, well known that the daily temperature amplitude is lower over dense vegetation compared to lower density or bare soil (Alkama & Cescatti, 2016; Duveiller *et al.*, 2018; Feldman *et al.*, 2019). Since many of the meteorological sites used in the development of this product occur in low-elevation, homogenous and developed areas, climatic similarity statistics are done using the WorldClim_v1.4 (Hijmans *et al.*, 2005) data that is mainly built to overcome these kind of problem by, for example, the use elevation statistics. In addition, in order to account for the climate drivers (eg. Solar radiation), the geographic weighted regression is used within the in-situ air temperature. Despite the fact this product uses some complexes statistics to overcome the issues described here and also the fact that this product is validated (Hooker *et al.*, 2018) against ERA5 reanalysis data, we cannot exclude that part of the uncertainties found in this study originates from the use of this product itself especially over elevated land where the difference with ERA5 is larger". This paragraph is included in the new version of the manuscript (see lines 420-434).

Based on our new analysis (see Figs 1 and 2 below), there is no fundamental difference between LST collection 5 and 6 in term of trend and inter-annual variability. We have to note here that our analysis are based on the inter-annual variability. In addition, the small difference between the two products, in principle, cannot impact our results for the three main reasons: 1) The systematic bias that may exist vanish by the use of dT (difference between two years) in our analysis; 2) By using observed in-situ air temperature to fit LST to air temperature, we somehow correct the potential bias that may exist in the collection 5; 3) the potential average bias that may exist inside the 50km radius distance is reduced when cleaning up the inter-annual variability from surrounding stable grid cells. These are, probably, the reasons why our analysis shows robust signal (Fig S9-S14) despite using different LAI products.

Alkama R, Cescatti A (2016) Climate change: Biophysical climate impacts of recent changes in global forest cover. *Science*, **351**, 600–604.

Duveiller G, Hooker J, Cescatti A (2018) The mark of vegetation change on Earth's surface energy balance. *Nature Communications*, **9**, 679.

Feldman AF, Short Gianotti DJ, Trigo IF, Salvucci GD, Entekhabi D (2019) Satellite-Based Assessment of

Land Surface Energy Partitioning–Soil Moisture Relationships and Effects of Confounding Variables. *Water Resources Research*, **55**, 10657–10677.

Hijmans RJ, Cameron SE, Parra JL, Jones PG, Jarvis A (2005) Very high resolution interpolated climate surfaces for global land areas. *International Journal of Climatology*, **25**, 1965–1978.

Hooker J, Duveiller G, Cescatti A (2018) A global dataset of air temperature derived from satellite remote sensing and weather stations. *Scientific Data*, **5**, 180246.

Figure 1 MODIS AQUA collection 5 and 6 $LST = (LST_{day} + LST_{night})/2$ trends over 2003-2014

Figure 2 Percentage of land grid-cells binned by classes of correlation of 0.02. This monthly correlation is computed between LST MODIS AQUA collection 5 and 6 over 2003-2014. On average, 67% of land grid cells shows a correlation greater than 0.98 and 91% of grid cells greater than 0.9.

3. LAI Product: The authors state that they evaluated 5 LAI products but do not explicitly state which ones (Line 322). They then go on to state, “First, all products, except GLASS, are produced using an ensemble of different satellites/sensors in time over 2003-2014, which may introduce inconsistencies in the time series⁴⁶” (Line 324). I take issue with this statement and with the way the author’s have evaluated LAI products. First, many products are based on only MODIS surface reflectance observations over the full 2003-2014 period. These are the best available observations over the full time period, and thus it seems most appropriate to limit the LAI product comparison to those that are derived from MODIS surface reflectance if possible. It is my understanding that GLASS and GLOBMAP LAI products are based on MODIS C5 surface reflectance (Jiang et al 2017). Jiang et al 2017 found that the MODIS C6 LAI trend is positive, whereas the GLASS and MODIS C5 LAI trend is negative from 2003-2011 (very similar to the period of study here). This difference is attributed to MODIS sensor degradation, which was corrected in the MODIS C6 product. How does uncertainty in LAI trends impact the findings presented here? I acknowledge that it’s very good to see generally consistent relationships between LAI and temperature (Figure S12). That said, I feels it’s important that the author’s are using sound logic in their LAI product evaluation and able to show their findings are robust across multiple LAI products.

Jiang C, Ryu Y, Fang H, Myneni R, Claverie M, Zhu Z. 2017. Inconsistencies of interannual variability and trends in long-term satellite leaf area index products. *Global Change Biology* 23: 4133–4146.

The name of the LAI products were included few lines below and are now also reported in such line (see line 325-326). In addition, as said before, we used GLASS LAI v40 that is based on **MODIS collection 6**. It is true that GLOBMAP is also based on MODIS over 2003-2014, thanks for the correction. This is now corrected in the method section line 328 *“First, all products, except GLASS and GLOBMAP, are produced using an ensemble of different satellites/sensors in time over 2003-2014, which may introduce inconsistencies in the time series⁴⁶”*. GLOBMAP was excluded because of its coarser resolution (0.073 instead of 0.05 deg). Indeed, in order to use GLOBMAP we have to re-grid Ts from 0.05 to 0.73 degree which may introduce uncertainties. This is the main reason why it was excluded in the previous version of the manuscript. Here, following the recommendations of the reviewer, we included GLOBMAP in the new version of the manuscript (see fig 3 and 4 bellow similar to S13 and S14 in the supplementary materials). As expected, GLOBMAP do not have a significant difference with COPERNICUS and GLASS but shows bigger spread which we suspect to be generated from the re-grid itself.

In addition, as clearly stated in the method section, we tested 5 LAI products over 2003-2004 and find a general consistent pattern (Fig S9-S12) between them despite their trend divergence over time. The signal is consistent because of using inter-annual variability (which is known to be much bigger than the trend) rather than trends themselves. For example, GLASS or GLOBMAP and COPERNICUS LAI shows consistent dT/dLAI (Figure 3-4 bellow, S13-S14 in the supplementary materials) despite the fact that they are originated from different sensors and experiencing different trends. This figure, demonstrate that the choice of observed LAI product has a very little impact on future projected temperature induced LAI change compared to the large spread between CMIP6 experiments.

Figure 3 Mean biophysical impact of LAI change on air temperature over 2015-2100 under the four SSPs, based on GLASS LAI in red, COPERNICUS in black and GLOBMAP in green.

Figure 4 Mean annual sensitivities derived from Earth observations by using all combinations of pairs of years in 2003-2014 for each month of the year using GLASS LAI on top, COPERNICUS in middle and GLOBMAP in bottom.

4. The loss of snow cover in winter at higher latitudes should result in local warming due to changes in the albedo. However, a main issue with passive satellite observations at high latitude in the winter is a lack of daylight hours and high solar zenith angles. Thus, how confident are the authors in their high-latitude, winter-time estimates of $dT/dLAI$? What steps were taken to account for these issues with satellite observations?

Good question. Indeed, in order to increase the accuracy of our estimates, we used all possible combinations between two years from 2003-2014 period which correspond to 66 samples. We then excluded grid cells with standard deviation of $dT/dLAI$, from the 66 samples, larger than 0.2. Despite that, figure 1d shows large spread in high latitudes variations across longitudes but the general patterns still similar to the two other products (GLOBMAP and COPERNICUS, Fig 4 above). This issue is discussed in the main text lines 203-210.

5. Again, very little detail is provided on the input environmental variables used in the analysis. For instance, a single line indicates that evaporation data from the GLEAM model was used in the analysis (Line 301). However, there are multiple GLEAM model versions and data products. There needs to be more detail included here so that the analysis is reproducible. Details could reveal issues with the analysis also. For instance, does this product use satellite LST data as an input? If so there is circularity when comparing changes in LST-based air temperature with LST-based estimates of transpiration.

Another good question, thanks. Indeed, we used GLEAM version 3.1a which uses Air temperature derived from measurements of the Atmospheric Infrared Sounder (AIRS, Aumann et al., 2003). MODIS LST is not used in this product, thus, in principle, there is no circularity problem with LST-based air temperature. The GLEAM version that is used in current analysis is clearly reported in the new version of the manuscript (see line 297-298).

Aumann, H. H., Chahine, M. T., Gautier, C., Goldberg, M. D., Kalnay, E., McMillin, L. M., Revercomb, H., Rosenkranz, P. W., Smith, W. L., Staelin, D. H., Strow, L. L., and Susskind, J.: AIRS/AMSU/HSB on the Aqua mission: design, science objectives, data products, and processing systems, *IEEE T. Geosci. Remote Sens.*, 41, 253–264, doi:10.1109/TGRS.2002.808356, 2003.

6. It is also unclear how sensitivity $dTs/dLAI$ was calculated "as a function" of snow cover, ET, and radiation (Lines 303 - 307). Were ET, radiation, and snow explicitly included as variables in a multivariate framework? Additional details are needed before the sensitivity analysis can be fully evaluated.

We realized that with the previous method section, it was not possible to fully understand all the steps used in this study. Indeed, we first, estimated $dTs/dLAI$ as described in the method section 1 for each of the 12 months across the 66 combination of two years. We then split the world on two, with and without snow cover (method section 2). In each of the two regions a bivariate quadratic least square regression is used. This is now well explained the new method section 2 that also includes the equations.

7. A final potentially major issue with this analysis is that there is no explicit treatment of land cover change. Previous research has found that for key regions much of the trend in LAI is driven by changes in land cover (Wang et al. 2019, 2020). Indeed, land cover change would result in a step change in LAI and LST that would greatly impact $dT/dLAI$ estimates. It is problematic if these estimates are then applied to extrapolate dT from $dLAI$ in ESMs (especially if these ESMs used fixed land cover maps), which are largely predicting vegetation density change. At minimum, this limitation of the approach needs to be discussed in detail.

Wang JA, Friedl MA. 2019. The role of land cover change in Arctic-Boreal greening and browning trends. *Environmental Research Letters* 14: 125007.

Wang JA, Sulla-Menashe D, Woodcock CE, Sonnentag O, Keeling RF, Friedl MA. 2020. Extensive land cover change across Arctic–Boreal Northwestern North America from disturbance and climate forcing. *Global Change Biology* 26: 807–822.

We agree on the fact that trends in LAI in some key regions are driven by the change in land cover. On one hand, our analysis are based on the inter-annual variability not on the trends. As known and said before, the inter-annual variability is much bigger than the trends in both stable and instable vegetation cover. It may (inter-annual variability), in some cases, have a bigger impact on LAI than land cover change. In addition, all ESMs includes land use change (the anthropogenic part of land cover change) as described by (Hurtt *et al.*, 2020) in both historical simulations and future scenarios. Wildfires and tree mortality are also simulated by ESMs. These are the main drivers of land cover change (Wang et al. 2020). However, in general, natural shifts in plant species due to climate change is not simulated. This limitation is discussed in the new version of the manuscript (see lines 210-212). On the other hand, by expressing $dT/dLAI$ as function of evaporation, snow cover and incoming solar radiation, we somehow account for the potential impact of land cover change on $dT/dLAI$ since land cover change impacts $dT/dLAI$ mainly via change in evaporation or snow cover.

Hurtt GC, Chini L, Sahajpal R et al. (2020) Harmonization of global land use change and management for the period 850–2100 (LUH2) for CMIP6. *Geoscientific Model Development*, **13**, 5425–5464.

8. Related to Comment 4, how do the models treat land cover change out to 2100? Are some using static land cover? With climate change, we know there could be rapid shifts in ecosystem composition from the conversion of tropical forests to savannas to the shrubification of the arctic. Given different vegetation function types have different $dT/dLAI$ due to different life history strategies, I suspect this is a large area of simplification in this study. This study is still very valuable and these complexities are not easily quantified, however, they should be stated as a limitation and discussed.

As we said before (see answer 7), all climate models use land cover change scenarios from land use harmonization datasets (Hurtt et al. 2020, <https://luh.umd.edu/>) and simulate tree mortality and fires. However, natural shifts in plant species is not simulated. This is discussed in the text (see lines 210-212

and lines 403-406). In contrary to the reviewer statement, we think that $dT/dLAI$ can be different even for the same vegetation function type and this difference is mainly due to the difference in snow cover, evaporation and solar radiation. However, for same evaporation, snow cover and solar radiation, we expect more or less same $dT/dLAI$ even for different land cover types.

9. Is it appropriate to extrapolate static T_s sensitivities found here to 2100? The sensitivity estimates are based on 10 years worth of data and it's unrealistic to assume they will remain static over time. We know ecosystems adapt to climate over long time periods. For example, trees might respond to increasing atmospheric CO_2 by modifying key hydraulic traits, which might lead to a change in ET and consequently a change in air temperature, and this could happen independent of change in LAI. There should at minimum be discussion of this simplifying assumption.

We completely agree with the reviewer on this. This is one of the reason why we expressed $dT/dLAI$ as function of evaporation. Future evaporation came from CMIP6 simulations which already accounts for the adaptation of plants, for example to increasing atmospheric CO_2 . In principle the effects described by the reviewer are accounted in our analysis. We already mentioned this in the method section (lines 250-253) but we agree with reviewer on the fact that we should also address this issue in the main text. This is what we did in the new version of the manuscript (see lines 106-109).

10. Additionally, I feel the authors should at least address the case that projected increases in LAI may be overstated in ESMs (Smith et al. 2016; Yuan et al. 2019; Wang et al. 2020). If this is the case and we are moving towards a warmer world with slowing LAI increases and increasing moisture constraint, could the authors infer from the observational work what might happen with biophysical climate regulation? Could there be a positive feedback in this case? It could be useful to point this potential option out in the discussion as a point of uncertainty and need for future research.

Wang, S., et al. 2020. Recent global decline of CO_2 fertilization effects on vegetation photosynthesis. *Science* 370, 1295-1300.

Yuan, W., Zheng, Y., Piao, S., Ciais, P., Lombardozzi, D., Wang, Y., Ryu, Y., Chen, G., Cox, P., Dong, W., Hu, Z., Jain, A.K., Jiang, C., Kato, E., Li, S., Lienert, S., Liu, S., Nabel, J., Qin, Z., Quine, T., Sitch, S., Smith, W.K., Wang, F., Wu, C., Xiao, Z., Yang, S. 2019. Increased atmospheric vapor pressure deficit reduces global vegetation growth. *Science Advances* 5, eaax1396.

Smith, W.K., Reed, S.C., Ballantyne A.P., Cleveland, C.C., Anderegg, W.R.L., Wieder W.R., Running, S.W. 2016. Large divergence of satellite and Earth system model estimates of global terrestrial CO_2 fertilization. *Nature Climate Change* 6, 306–310.

Thanks for this comment and for the references. For the reason described by the reviewer, we expect slowing down of the LAI increase in the future compared to what shows large number of ESMs. But, we expect at least few models to be in good shape. We think, we were as realistic as possible in our analysis because of showing the results coming from 18 ESMs over 4 different SSPs. We hope to catch the truth between the large spread shown by climate models. This limitation is discussed in the text (see lines 213-217).

Minor Comments:

1. Abstract: SSps and CMIP5 are undefined acronyms.

Removed from the text because Nat Com do not allow the use of acronyms in the abstract and we are limited to a maximum of 150 words.

2. Line 301. GLEAM undefined.

Ok, done (see line 297-298)

3. Line 322: State the five products that were compared up front.

Ok, done (see lines 325-326)

4. Why are Fig. 1a,b cut off at SWdown 350 W/m²?

As mentioned in the method section, we used ERA5 monthly SWdown which is in general lower than 350 W/m² (Figure 5) but we agree on the fact that in some years and particular months we may reach 380 W/m². In the new version of the manuscript we extended the figure to 380w/m². In some cases, it may also overpass the 380 w/m² but, from our tests, this only happen over un-vegetated areas.

Figure 5 Monthly mean incoming solar radiation at the surface over 2003-20014

Reviewer #2 (Remarks to the Author):

Reviews of Manuscript No.: NCOMMS-21-18083-T

Title: Vegetation-based climate mitigation in a warmer and greener World

Author(s): Ramdane Alkama, Giovanni Forzieri, Gregory Duveiller, Giacomo Grassi, Shunlin Liang and Alessandro Cescatti

Overall conclusions and recommendations:

This manuscript tackles an original question regarding the mitigation potential of vegetation-driven biophysical effects under different future climate change scenarios. Authors use remote sensing datasets including air temperature and LAI to estimate the temperature effects in LAI changes. According to the temperature sensitivity, authors further establish a function to explore the temperature sensitivity effected by snow cover, solar radiation and evaporation fluxes. Authors then calculated the temperature changes induced by the LAI altering under four climate change scenarios based on the CMIP6 archive. The main findings including 1) the biochemical effect shows more important than the biophysical effect in the greening world, 2) a large potential of vegetation to reduce future land warming in the most ambitious mitigation plan SSP126 scenario than the business as usual SSP585 scenario.

The study is within the scope of the journal Nature Communications, and its topic and conclusions could attract the interest of readers working in areas of climate change mitigation and land use/cover management. This manuscript has a clear structure, the quality of presentation is acceptable. It has potential to be a good reference. In the following, I provide more specific comments (major and minor) and suggestions to improve the manuscript.

We thank the reviewer for the positive comments.

Specific comments:

1. Authors introduce a function between temperature changes induced by LAI change and three environmental drivers, solar radiation, evaporation and snow cover. However the function is not clear. Although the manuscript shows the function in Figure 1ab. But it not easy to follow. Please add the equations.

Thanks for this comment. Indeed, this part was missing in the previous version. This is now included in the method section (See the new method section 2).

2. The manuscript distinguishes the biophysical and biochemical effect of LAI on air temperature. How to distinguish them?

The biochemical effect is the fact that plants absorb some of the CO₂ emitted by fossil fuels. This part is estimated from the use of **total carbon stock in the plants** (not only leaves) as simulated by CMIP6 simulations. LeDuc et al. 2016 find a linear relationship between carbon in the atmosphere and air surface temperature. We used this linear equation as well as the increase in carbon stock in plants (difference between 2100 and 2015) for each model to find the biochemical effect of the greening on surface temperature (see method and data section 6 and main text: lines 224-226).

The biophysical effect is estimated from comparing nearby grid cells where we observe change in LAI compared to stable grid cells after cleaning up the inter-annual variability from the signal. Indeed, the change in plant density may influence air temperature mainly through the change in evaporation or snow cover (this is documented in the introduction paragraph 2, 3 and 4 and method and data sections 1 and 2).

3. This manuscript introduces multi-datasets including remote sensing datasets, climate modelling datasets from CMIP6 archive. Different data/model have different spatial resolution. How to compare these datasets? Which resolution/grid size is used in the analysis?

The remote sensing datasets are used at 0.05 degree spatial resolution in order to estimate the observed sensitivity $dt/dLai$ as function of evaporation, solar radiation and snow cover. Future $dT/dLai$ is estimated at 2 degree spatial resolution using the observed function and CMIP6 climate outputs re-gridded into 2x2 degree spatial resolution. This is now reported in the manuscript (see lines 246-250)

4. "Results show that, under the SSP585 scenario, by 2100 greening will likely mitigate land warming by $0.71 \pm 0.40^\circ\text{C}$, and 83% of such effect ($0.59 \pm 0.41^\circ\text{C}$) is driven by the increase in plant carbon sequestration, while the remaining cooling ($0.12 \pm 0.05^\circ\text{C}$) is due to biophysical land-atmosphere interactions." I can not find/read these numbers in any figures.

Figure 5abc shows the median, 1st and 3rd quartile of mean land warming, total biophysical and biochemical mitigation estimates from the 14 climate models, while the reported numbers in the abstract are for the average \pm the standard deviation. These numbers are now reported in Table 1 in the supplementary materials.

5. L25, "Half of the biophysical mitigation effect ...", how to quantify the half?

This is estimated from figure 4a,b,c. In the new version of the manuscript, we included a paragraph that explain how this figure 4 is generated (see lines 435-443).

6. L32-37, the sentence is not easy to follow, please revise it.

This sentence is deleted from the new version of the manuscript.

7. L140, and SSP245 (Fig. S2gh) -> and SSP245 (Fig. S3gh)

Ok, thanks

8. L300, add reference for ERA5 reanalysis.

Ok, done (see line 292).

9. Figure 5b, y-axis title, Biogeochemical -> Biochemical

Figure 5 is not being cited in the manuscript. Please check it.

Ok, corrected (see new figure 5). This figure, is cited in the new version of the manuscript (see lines 201, 226-237).

Reviewer #3 (Remarks to the Author):

The study by Alkama et al. investigated the biophysical and biochemical impacts of recent and future greening on temperature using observations and CMIP6 simulations. The authors suggested that the Earth's greening can be a potential approach for future climate mitigation due to its cooling effects. Overall, this study is on a topic of relevance and general interest to the readers of Nature Communication. However, the methodology needs some clarification, and the results need more explanations and discussions.

We thank the reviewer for his positive comments.

My major concerns are:

1. It is not clear how $dT/dLAI$ is calculated. For instance, it is very confusing to see $dT/dLAI$ and $dT/d\ln(LAI)$ in Figure 1. What is their difference? Regarding Method section 1.2, what is the reason to use snow cover, solar radiation, and ET to describe climate conditions? Why not use temperature and precipitation? "The sensitivity is estimated as a function of monthly snow cover and solar radiation", "the sensitivity is formulated as a function of land evaporation fluxes and solar radiation". Can the authors provide more explanations of their methods here?

We did not use temperature and precipitation because it is well known that the biophysical cooling/warming effect of vegetation to be mainly originated from the change in snow cover over snowy regions and from the change in evaporation elsewhere. This the main reason why we used "snow cover" and "evaporation" in the sensitivity function. Solar radiation is also important since it plays an important role for determination the net radiation at the surface. It also allow us to represent the seasonality and then use one single function instead of one function for each month of the year. These arguments are reported in the text (see lines 69-83).

2. I like the idea of separating the biophysical and biochemical impacts. However, without reading the paper (Leduc et al 2016), readers cannot understand how the biochemical effects are calculated? As the authors suggested, the biogeochemical feedback of Earth's green is about five times larger than the biophysical feedback. However, there is no evaluation of the simulated biogeochemical feedback in CMIP6. For instance, can the biochemical effects in CMIP6 be compared with observations like the biophysical effects in Figure 1? Results in Figure 5 are never explained or discussed in the manuscript.

In the new version of the manuscript the method behind the biochemical effect is well explain and figure 5 is discussed (see lines 224-237 and lines 408-418).

Concerning the evaluation of the biochemical effect, all previous IPCC reports (eg.

<https://www.ipcc.ch/site/assets/uploads/2018/02/ar4-wg1-chapter8-1.pdf>,

https://science2017.globalchange.gov/downloads/CSSR_Ch3_Detection_and_Attribution.pdf) showed that ESMs are able to simulate correctly observed trend in air temperature and such trend is mainly driven by increased atmospheric CO₂. On the other hand, Leduc et al. 2016, using CMIP5 ESMs, shows a linear relationship between atmospheric CO₂ and surface temperature. Here we used this linear relation and the amount of carbon that is stored by plant, coming from CMIP6 models, to estimate the biochemical effect of plants on air temperature. Basically, Leduc et al. 2016 find an increase of land temperature by $2.2 \pm 0.5^\circ$ per 1 Terra ton of carbon (Tt C) in the atmosphere. In our case we used the total increase of carbon in plants (ΔB in Tt C) between 2015 and 2100 coming from CMIP6 archive to estimate the biochemical effect as

$$\Delta T = \frac{2.2 \Delta B}{1}$$

3. In Figure 4, how were the “all effect”, “climate change effect (and its individual terms)”, “LAI effect” defined and calculated? In the “climate change effect”, would the “Evap” effect be considered as the results of the “LAI effect”?

“LAI effect” is estimated from the multiplication of current sensitivity $dT/dLai$ by Lai trend. In “all effects”, we first estimate new $dT/dLai$ from simulated future solar radiation, evaporation and snow cover, and then multiplied the new sensitivity by LAI change. While, “climate change” effect is the difference between the two. Concerning the individual terms of climate change are estimated by subtracting “all effects” from “all effect except individual term that was kept constant”. This explanation is included in the new version of manuscript (see methods section 8).

Yes, “Evap” is the result of LAI effect that included adaptation to future climate (using future $dT/dLAI$) over snow free areas.

Others:

1. It would be better to use a consistent term: “evaporation” vs. “evapotranspiration”; “biogeochemical” vs. “biochemical”; “biogeophysical” vs. “biophysical”.

Ok, done.

2. According to Figure 2, the sensitivity ($dT/dLAI$) is getting weaker and ET is getting stronger in India, parts of Africa and South America, where there is high sensitivity (more negative) at present according to Figure 1. First, the statement “the increased sensitivity is mostly observed in the regions where evaporation is increasing” seems incorrect. It is increased $dT/dLAI$, but decreased sensitivity. Second, any explanation about what caused the weaker sensitivity?

Thanks for this relevant comment. Yes, it is increasing $dT/dLai$ but decreased sensitivity (this is now corrected in the new version of the manuscript). The explanation of this can be raised from Fig 1 which

shows clearly that areas with high evaporation like Amazonian forest experience lower sensitivity compared to dry zone. Therefore, increased evaporation in such area lead to decreased sensitivity. This is now corrected (see lines 158-162).

We thank the reviewers for their constructive comments that helped us improving the manuscript.

REVIEWER COMMENTS

Reviewer #1 (Remarks to the Author):

This study aims to quantify surface air temperature (T_s) changes induced by future leaf area index (LAI) dynamics derived from four Shared Socioeconomic Pathways (SSPs) from an ensemble of Coupled Model Intercomparison Project 6 (CMIP6) Earth System Models (ESMs). In response to widespread projected increases in LAI across all models, all scenarios show a progressive biophysical cooling that is positively correlated with greening. This is an interesting paper on an important mechanism that is missing from ESMs. The research is well executed, and the writing is compelling. The work is also of broad importance given that it provides an estimate of the potential impact of biophysical climate regulation associated with changes in LAI.

I reviewed a previous version of this manuscript. I find that the authors have done a commendable job in addressing all my initial concerns. In particular, I find that the research methods have been more clearly described, and I now feel convinced of their robustness, and importantly their reproducibility. I recommend this manuscript for publication in Nature Communications.

Reviewer #2 (Remarks to the Author):

All my concerns have been addressed in the revisions. I give a high recommendation of this manuscript.

Reference 2 and 5 have not been cited in the manuscript. Please add the citation or delete the two references.

A typo error for y-axis title of Figure 5b. Biochiminal ->Biochemical

Reviewer #3 (Remarks to the Author):

I appreciate the efforts that the authors have made in response to my questions and concerns. However, there are still some issues.

1. The authors provided more explanation of how to calculate the biochemical effect of LAI on air temperature. I have two questions here. First, the sensitivity of temperature to land carbon in Leduc (2016) is derived from 12 CMIP5 models. We know that there are many changes (and improvements) from CMIP5 models to CMIP6 models. Can we directly apply this value (2.2) to the 18 CMIP6 modes in this study? Second, Equation (6) only tells the temperature response to the increased land carbon. How to connect temperature, land carbon, and LAI in their analysis? How do we know how much of the land-carbon-induced temperature change is due to LAI change?

2. The authors may misunderstand my previous comment about Figure 2. My concern is - in India, parts of Africa and South America, the sensitivity is getting weaker (Figure 2i), but the ET is getting stronger (Figure 2c) in those regions. Therefore, spatially speaking, the increased

sensitivity is mostly observed in the mid and high latitudes where ET is increasing. Also, the statement "the decreased sensitivity is linked with the reduction in snow cover and/or evaporation" is inaccurate. Decreased temperature sensitivity is found in India, parts of Africa and South America, where ET is actually increasing and there is no change in snow. So changes in ET and snow cover may not be the reasons for the decreased sensitivity in those regions.

3. I feel the authors finished the revisions in a rush. There are many grammar issues in the added sentences/paragraphs. They should carefully read the manuscript before resubmitting the revisions. Some examples are:

L77: we have to not here -> we have to note here

L263: as function -> as a function

L267: both equation -> both equations

L267-269: this sentence is confusing. Please revise.

L273: in this regions -> in those regions

L362: coming from CMIP6 archive -> from the CMIP6 archive

L363: by the equation (6) -> by equation 6

L370: One of the reason of the use -> One of the reasons for the use

L376: these kind of -> this kind of

L377: geographic weighted -> geographically weighted

L378: some complexes statistics -> some complex statistics

AUTHOR'S RESPONSE TO REVIEWER'S COMMENTS

REVIEWER COMMENTS

Reviewer #1 (Remarks to the Author):

This study aims to quantify surface air temperature (T_s) changes induced by future leaf area index (LAI) dynamics derived from four Shared Socioeconomic Pathways (SSPs) from an ensemble of Coupled Model Intercomparison Project 6 (CMIP6) Earth System Models (ESMs). In response to widespread projected increases in LAI across all models, all scenarios show a progressive biophysical cooling that is positively correlated with greening. This is an interesting paper on an important mechanism that is missing from ESMs. The research is well executed, and the writing is compelling. The work is also of broad importance given that it provides an estimate of the potential impact of biophysical climate regulation associated with changes in LAI.

I reviewed a previous version of this manuscript. I find that the authors have done a commendable job in addressing all my initial concerns. In particular, I find that the research methods have been more clearly described, and I now feel convinced of their robustness, and importantly their reproducibility. I recommend this manuscript for publication in Nature Communications.

Thanks for your time.

Reviewer #2 (Remarks to the Author):

All my concerns have been addressed in the revisions. I give a high recommendation of this manuscript.

Reference 2 and 5 have not been cited in the manuscript. Please add the citation or delete the two references.

A typo error for y-axis title of Figure 5b. Biochimical ->Biochemical

Ok, done see new figure 5. Thanks for pointing this.

Reviewer #3 (Remarks to the Author):

I appreciate the efforts that the authors have made in response to my questions and concerns. However, there are still some issues.

1. The authors provided more explanation of how to calculate the biochemical effect of LAI on air temperature. I have two questions here. First, the sensitivity of temperature to land carbon in Leduc (2016) is derived from 12 CMIP5 models. We know that there are many changes (and improvements) from CMIP5 models to CMIP6 models. Can we directly apply this value (2.2) to the 18 CMIP6 models in this study? Second, Equation (6) only tells the temperature response to the increased land carbon. How to connect temperature, land carbon, and LAI in their analysis? How do we know how much of the land-carbon-induced temperature change is due to LAI change?

In the historical simulations from both CMIP5 and CMIP6, climate models simulate reasonably well global warming since pre-industrial (IPCC 2021) and such warming is mainly due to the increases in greenhouse gases (mainly CO₂). For this reason, we expect to find a similar relationship between atmospheric CO₂ and land temperature in both CMIP5 and CMIP6 experiments. For example, using CMIP5 simulations, Leduc (2016) find 2.2 ± 0.5 which is in agreement with observations 2.3 ± 0.1 (see figure 1 below). In this graph, we used temperature coming from CRU.ts4.05 and observed atmospheric CO₂ from (Meinshausen *et al.*, 2017).

Plants can mitigate climate change by two main mechanisms: the biophysical and the biochemical one. The **biophysical** effects are mainly driven by the interactions between **leaves (LAI)** and the atmosphere, while **biochemical** effects depend mostly in changes in **vegetation biomass, and mainly wood biomass**. This is the reason why we used LAI to study the biophysical effect, while we use total vegetation carbon instead to assess the biochemical effect. We see no particular reason to quantify the biochemical effect of LAI, which should be marginal compared to that of woody biomass i.e. the largest vegetation carbon stock. We included a sentence to clarify this difference in the current version of the manuscript. See lines (204-207).

IPCC, 2021: Climate Change 2021: The Physical Science Basis. Contribution of Working Group I to the Sixth Assessment Report of the Intergovernmental Panel on Climate Change [Masson-Delmotte, V., P. Zhai, A. Pirani, S. L. Connors, C. Péan, S. Berger, N. Caud, Y. Chen, L. Goldfarb, M. I. Gomis, M. Huang, K. Leitzell, E. Lonnoy, J. B. R. Matthews, T. K. Maycock, T. Waterfield, O. Yelekçi, R. Yu and B. Zhou (eds.)]. Cambridge University Press. In Press.

Meinshausen M, Vogel E, Nauels A et al. (2017) Historical greenhouse gas concentrations for climate modelling (CMIP6). *Geoscientific Model Development*, **10**, 2057–2116.

Figure 1 Overall land temperature (CRU.ts4.05) responses to cumulative CO₂ emissions over 1901-2020. Each dot represent one year. The mean observed response over land is 2.3 ± 0.1 °C per TtC while the estimated response from CMIP5 simulations (Leduc et al. 2016) is about 2.2 ± 0.5 °C per TtC (small right plot).

2. The authors may misunderstand my previous comment about Figure 2. My concern is - in India, parts of Africa and South America, the sensitivity is getting weaker (Figure 2i), but the ET is getting stronger (Figure 2c) in those regions. Therefore, spatially speaking, the increased sensitivity is mostly observed in the mid and high latitudes where ET is increasing. Also, the statement “the decreased sensitivity is linked with the reduction in snow cover and/or evaporation” is inaccurate. Decreased temperature sensitivity is found in India, parts of Africa and South America, where ET is actually increasing and there is no change in snow. So changes in ET and snow cover may not be the reasons for the decreased sensitivity in those regions.

Thanks a lot for pointing this. We completely agree. Thus, accordingly, we made the necessary changes in the main text as follow “As a consequence of future change in background climate conditions, the variation in $dT/dLAI$ sensitivity show a larger change in the boreal zone (Fig. 2ij). Indeed, the large decrease in $dT/dLAI$ is linked to the reduction in snow cover. Elsewhere, the trend in the sensitivity is more complex. For example, Fig 1 shows clearly that areas with higher evaporation, such as the tropical

rainforests in the Amazonian basin, experience lower $dT/dLAI$ compared to vegetation in a dry climate. This may explain why in India, parts of Africa and South America, the sensitivity is getting weaker (Figure 2i), but the evaporation is getting stronger.” (see lines 127-134)

3. I feel the authors finished the revisions in a rush. There are many grammar issues in the added sentences/paragraphs. They should carefully read the manuscript before resubmitting the revisions. Some examples are:

L77: we have to not here -> we have to note here

L263: as function -> as a function

L267: both equation -> both equations

L267-269: this sentence is confusing. Please revise.

L273: in this regions -> in those regions

L362: coming from CMIP6 archive -> from the CMIP6 archive

L363: by the equation (6) -> by equation 6

L370: One of the reason of the use -> One of the reasons for the use

L376: these kind of -> this kind of

L377: geographic weighted -> geographically weighted

L378: some complexes statistics -> some complex statistics

Ok done, thanks.

We thank the reviewers for their comments that helped us to improve the manuscript.

REVIEWERS' COMMENTS

Reviewer #3 (Remarks to the Author):

All my concerns have been well addressed in the current version of the manuscript. I appreciate the efforts that have been made by the authors.